# DIFFERENTIALLY PRIVATE $L_2$-HEAVY HITTERS IN THE SLIDING WINDOW MODEL

**Jeremiah Blocki, Seunghoon Lee & Tamalika Mukherjee**
Purdue University
West Lafayette, IN 47906, USA
{jblocki,lee2856,tmukherj}@purdue.edu

**Samson Zhou**
UC Berkeley and Rice University
Houston, TX 77005, USA
samsonzhou@gmail.com

## ABSTRACT

The data management of large companies often prioritize more recent data, as a source of higher accuracy prediction than outdated data. For example, the Facebook data policy retains user search histories for 6 months while the Google data retention policy states that browser information may be stored for up to 9 months. These policies are captured by the sliding window model, in which only the most recent $W$ statistics form the underlying dataset. In this paper, we consider the problem of privately releasing the $L_2$-heavy hitters in the sliding window model, which include $L_p$-heavy hitters for $p \leq 2$ and in some sense are the strongest possible guarantees that can be achieved using polylogarithmic space, but cannot be handled by existing techniques due to the sub-additivity of the $L_2$ norm. Moreover, existing non-private sliding window algorithms use the smooth histogram framework, which has high sensitivity. To overcome these barriers, we introduce the first differentially private algorithm for $L_2$-heavy hitters in the sliding window model by initiating a number of $L_2$-heavy hitter algorithms across the stream with significantly lower threshold. Similarly, we augment the algorithms with an approximate frequency tracking algorithm with significantly higher accuracy. We then use smooth sensitivity and statistical distance arguments to show that we can add noise proportional to an estimation of the $L_2$ norm. To the best of our knowledge, our techniques are the first to privately release statistics that are related to a sub-additive function in the sliding window model, and may be of independent interest to future differentially private algorithmic design in the sliding window model.

## 1 INTRODUCTION

Differential privacy (Dwork, 2006; Dwork et al., 2016) has emerged as the standard for privacy in both the research and industrial communities. For example, Google Chrome uses RAPPOR (Erlingsson et al., 2014) to collect user statistics such as the default homepage of the browser or the default search engine, etc., Samsung proposed a similar mechanism to collect numerical answers such as the time of usage and battery volume (Nguyên et al., 2016), and Apple uses a differentially private method (Greenberg, 2016) to generate predictions of spellings.

The age of collected data can significantly impact its relevance to predicting future patterns, as the behavior of groups or individuals may significantly change over time due to either cyclical, temporary, or permanent change. Indeed, recent data is often a more accurate predictor than older data across multiple sources of big data, such as stock markets or Census data, a concept which is often reflected through the data management of large companies. For example, the Facebook data policy (Facebook) retains user search histories for 6 months, the Apple differential privacy (Upadhyay, 2019) states that collected data is retained for 3 months, the Google data retention policy states that browser information may be stored for up to 9 months (Google), and more generally, large data collection agencies often perform analysis and release statistics on time-bounded data. However, since large data collection agencies often manage highly sensitive data, the statistics must be released in a way that does not compromise privacy. Thus in this paper, we study the (event-level) differentially private release of statistics of time-bounded data that only use space sublinear in the size of the data.

**Definition 1.1** (Differential privacy (Dwork et al., 2016))**.** *Given $\varepsilon > 0$ and $\delta \in (0,1)$, a randomized algorithm $\mathcal{A}$ operating on datastreams is $(\varepsilon, \delta)$-differentially private if, for every pair of*

*neighboring datasets $\mathfrak{S}$ and $\mathfrak{S}'$ and for all sets $E$ of possible outputs, we have,* $\mathbf{Pr}\left[\mathcal{A}(\mathfrak{S}) \in E\right] \leq e^{\varepsilon} \cdot \mathbf{Pr}\left[\mathcal{A}(\mathfrak{S}') \in E\right] + \delta$.

In the popular streaming model of computation, elements of an underlying dataset arrive one-by-one but the entire dataset is considered too large to store; thus algorithms are restricted to using space sublinear in the size of the data. Although the streaming model provides a theoretical means to handle big data and has been studied thoroughly for applications in privacy-preserving data analysis, e.g., (Mir et al., 2011; Blocki et al., 2012; Joseph et al., 2020; Huang et al., 2022; Dinur et al., 2023) and adaptive data analysis, e.g., (Avdiukhin et al., 2019; Ben-Eliezer et al., 2022b; Hassidim et al., 2020; Braverman et al., 2021a; Chakrabarti et al., 2022; Ajtai et al., 2022; Beimel et al., 2022; Ben-Eliezer et al., 2022a; Attias et al., 2023), it does not properly capture the ability to prioritize more recent data, which is a desirable quality for data summarization. The time decay model (Cohen & Strauss, 2006; Kopelowitz & Porat, 2008; Su et al., 2018; Braverman et al., 2019) emphasizes more recent data by assigning a polynomially decaying or exponentially decaying weight to "older" data points, but these functions cannot capture the zero-one property when data older than a certain age is completely deleted.

**The sliding window model.** By contrast, the *sliding window model* takes a large data stream as an input and only focuses on the updates past a certain point in time by implicitly defining the underlying dataset through the most recent $W$ updates of the stream, where $W > 0$ is the window parameter. Specifically, given a stream $u_1, \ldots, u_m$ such that $u_i \in [n]$ for all $i \in [m]$ and a parameter $W > 0$ that we assume satisfies $W \leq m$ without loss of generality, the underlying dataset is a frequency vector $f \in \mathbb{R}^n$ induced by the last $W$ updates of the stream $u_{m-W+1}, \ldots, u_m$ so that $f_k = |\{i : u_i = k\}|$, for all $k \in [n]$. Then the goal is to output a private approximation to the frequency $f_k$ of each heavy-hitter, i.e., the indices $k \in [n]$ for which $f_k \geq \alpha L_p(f)$, which denotes the $L_p$ norm of $f$ for a parameter $p \geq 1$, $L_p(f) = \|f\|_p = \left(\sum_{i=1}^{n} f_i^p\right)^{1/p}$.

In this case, we say that streams $\mathfrak{S}$ and $\mathfrak{S}'$ are neighboring if there exists a single update $i \in [m]$ such that $u_i \neq u_i'$, where $u_1, \ldots, u_m$ are the updates of $\mathfrak{S}$ and $u_1', \ldots, u_m'$ are the updates of $\mathfrak{S}'$.

Note that if $k$ is an $L_1$-heavy hitter, i.e., a heavy-hitter with respect to $L_1(f)$, then $f_k \geq \alpha L_1(f)$ so that $f_k \geq \alpha \left(\sum_{i=1}^{n} f_i\right) \geq \alpha \left(\sum_{i=1}^{n} f_i^2\right)^{1/2}$, and $k$ is also an $L_2$-heavy hitter. Thus, any $L_2$-heavy hitter algorithm will also report the $L_1$-heavy hitters, but the converse is not always true. Indeed, for the Yahoo! password frequency corpus (Blocki et al., 2016) ($n \approx 70$ million) with heavy-hitter threshold $\alpha = \frac{1}{500}$ there were $3,972$ $L_2$-heavy hitters, but only one $L_1$-heavy hitter. On the other hand, finding $L_p$-heavy hitters for $p > 2$ requires $\Omega(n^{1-2/p})$ space (Chakrabarti et al., 2003; Bar-Yossef et al., 2004), so in some sense, the $L_2$-heavy hitters are the best we can hope to find using polylogarithmic space. Although there is a large and active line of work in the sliding window model (Datar et al., 2002; Braverman & Ostrovsky, 2007; Braverman et al., 2014; 2016; 2018; 2020; Borassi et al., 2020; Woodruff & Zhou, 2021; Braverman et al., 2021b; Jayaram et al., 2022), there is surprisingly little work in the sliding window model that considers differential privacy (Upadhyay, 2019; Upadhyay & Upadhyay, 2021).

## 1.1 OUR CONTRIBUTIONS

In this paper, we consider the problem of privately releasing approximate frequencies for the heavy-hitters in a dataset defined by the sliding window model. We give the first differentially private algorithm for approximating the frequencies of the $L_2$-heavy hitters in the sliding window model.

**Theorem 1.2.** *For any $\alpha \in (0, 1), c > 0$, window parameter $W$ on a stream of length $m$ that induces a frequency vector $f \in \mathbb{R}^n$ in the sliding window model, and privacy parameter $\varepsilon > \frac{1000 \log m}{\alpha^3 \sqrt{W}}$, there exists an algorithm such that:*

(1) *(Privacy) The algorithm is $(\varepsilon, \delta)$-differentially private for $\delta = \frac{1}{m^c}$.*

(2) *(Heavy-hitters) With probability at least $1 - \frac{1}{m^c}$, the algorithm outputs a list $\mathcal{L}$ such that $k \in \mathcal{L}$ for each $k \in [n]$ with $f_k \geq \alpha \, L_2(f)$ and $j \notin \mathcal{L}$ for each $j \in [n]$ with $f_j \leq \frac{\alpha}{2} \, L_2(f)$.*

(3) *(Accuracy) With probability at least $1 - \frac{1}{m^c}$, we simultaneously have $|f_k - \widetilde{f_k}| \leq \frac{\alpha}{4} \, L_2(f)$ for all $k \in \mathcal{L}$, where $\widetilde{f_k}$ denotes the noisy approximation of $f_k$ output by the algorithm.*

*(4) (Complexity) The algorithm uses $\mathcal{O}\left(\frac{\log^7 m}{\alpha^6 \eta^4}\right)$ bits of space and $\mathcal{O}\left(\frac{\log^4 m}{\alpha^3 \eta^4}\right)$ operations per update where $\eta = \max\{1, \varepsilon\}$.*

**Pure differential privacy for $L_1$-heavy hitters.** Along the way, we develop techniques for handling differentially private algorithms in the sliding window model that may be of independent interest. In particular, we also use our techniques to obtain an $L_1$-heavy hitter algorithm for the sliding window model that guarantees *pure* differential privacy.

**Continual release for $L_2$-heavy hitters.** Finally, we give an algorithm for continual release of $L_1$ and $L_2$-heavy hitters in the sliding window model that has additive error $\frac{\alpha\sqrt{W}}{2}$ for each estimated heavy-hitter frequency and preserves pure differential privacy, building on a line of work (Chan et al., 2012; Upadhyay, 2019; Huang et al., 2022) for continual release. By comparison, the algorithm of (Upadhyay, 2019) guarantees $\mathcal{O}\left(W^{3/4}\right)$ additive error while the algorithm of (Huang et al., 2022) gives $(\varepsilon, \delta)$-differential privacy. We remark that since $\sqrt{W} \leq L_2(t - W + 1 : t)$ for any $t \in [m]$, where $L_2(t - W + 1 : t)$ denotes the $L_2$ norm of the sliding window between times $t - W + 1$ and $t$, then our improvements over (Upadhyay, 2019) for the continual release of $L_1$-heavy hitters actually also resolve the problem of continual release of $L_2$-heavy hitters. Nevertheless, the approach is somewhat standard and thus we defer discussion to the appendix.

## 1.2 RELATED WORK

**Dynamic structures vs. linear sketching.** Non-private algorithms in the streaming model generally follow one of two main approaches. The first main approach is the transformation from static data structures to dynamic structures using the framework of (Bentley & Saxe, 1980). Although the approach has been a useful tool for many applications (Dwork et al., 2010; Chan et al., 2011; 2012; Larsen et al., 2020), it does provide a mechanism to handle the implicit deletion of updates induced by the sliding window model. The second main approach is the use of linear sketching (Blocki et al., 2012; Bassily & Smith, 2015; Bun et al., 2019; Bassily et al., 2020; Huang et al., 2022), where the data $x$ is multiplied by a random matrix $A$ to create a small-space "sketch" $Ax$ of the original dataset. Note that sampling can fall under the umbrella of linear sketching in the case where the random matrix only contains a single one as the nonzero entry in each row. Unfortunately, linear sketching again cannot handle the implicit deletions of the sliding window model, since it is not entirely clear how to "undo" the effect of each expired element in the linear sketch $Ax$.

**Adapting insertion-only streaming algorithms to the sliding window model.** Algorithms for the sliding window model are often adapted from the insertion-only streaming model through either the exponential histogram framework (Datar et al., 2002) or its generalization, the smooth histogram framework (Braverman & Ostrovsky, 2007). These frameworks transform streaming algorithms for either an additive function (in the case of exponential histograms) or a smooth function (in the case of smooth histograms) into sliding window algorithms by maintaining a logarithmic number of instances of the streaming algorithm, starting at various timestamps during the stream. Informally, a function is smooth if once a suffix of a data stream becomes a $(1 + \beta)$-approximation of the entire data stream for the function, then the suffix is always a $(1 + \alpha)$-approximation, regardless of the subsequent updates in the stream. Thus at the end of the stream of say length $m$, two of the timestamps must "sandwich" the beginning of the window, i.e., there exists timestamps $t_1$ and $t_2$ such that $t_1 \leq m - W + 1 < t_2$. The main point of the smooth histogram is that the streaming algorithm starting at time $t_1$ must output a value that is a good approximation of the function on the sliding window due to the smoothness of the function. Therefore, the smooth histogram is a cornerstone of algorithmic design in the sliding window model and handles many interesting functions, such as $L_p$ norm estimation (and in particular the sum), longest increasing subsequence, geometric mean, distinct elements estimation, and counting the frequency of a specific item.

On the other hand, there remain interesting functions that are not smooth, such as clustering (Braverman et al., 2016; Borassi et al., 2020; Epasto et al., 2022), submodular optimization (Chen et al., 2016; Epasto et al., 2017), sampling (Jayaram et al., 2022), regression and low-rank approximation (Braverman et al., 2020; Upadhyay & Upadhyay, 2021), and crucially for our purposes, heavy hitters (Braverman et al., 2014; 2018; Upadhyay, 2019; Woodruff & Zhou, 2021). These problems cannot be handled by the smooth histogram framework and thus for these problems, sliding windows algorithms were developed utilizing the specific properties of the objective functions.

**Previous work in the DP setting.** The work most related to the subject of our study is (Upadhyay, 2019) who proposed the study of differentially private $L_1$-heavy hitter algorithms in the sliding window. Although (Upadhyay, 2019) gave a continual release algorithm, which was later improved by (Huang et al., 2022), the central focus of our work is the "one-shot" setting, where the algorithm releases a single set of statistics at the end of the stream, because permitting a single interaction with the data structure can often achieve better guarantees for both the space complexity and the utility of the algorithm. Indeed, in this paper we present $L_2$-heavy hitter algorithms for both the continual release and the one-shot settings, but the space/accuracy tradeoffs in the latter are much better than the former. (Upadhyay, 2019) also proposed a "one-shot" algorithm, which empirically performs well, but lacks the theoretical guarantees claimed in the paper, i.e., see Section 1.3.

Privately releasing heavy-hitters in other big data models has also received significant attention. (Dwork et al., 2010) introduced the problem of $L_1$-heavy hitters and other problems in the *pan-privacy* streaming model, where the goal is to preserves differential privacy even if the internal memory of the algorithm is compromised, while (Chan et al., 2012) considered the problem of continually releasing $L_1$-heavy hitters in a stream. The heavy-hitter problem has also been extensively studied in the local model (Bassily & Smith, 2015; Ding et al., 2017; Acharya & Sun, 2019; Bun et al., 2019; Bassily et al., 2020), where individual users locally add privacy to their data, e.g., through randomized response, before sending their private information to a central and possibly untrusted server to aggregate the statistics across all users.

## 1.3 Overview of Our Techniques

In this section, we give a brief overview of our techniques and the various challenges that they overcome. We defer full proofs to the supplementary material. We first use the smooth histogram to obtain a constant factor approximation to the $L_2$ norm of the sliding window similar to existing heavy-hitter non-DP algorithms in the sliding window model (Braverman et al., 2014; 2018). We maintain a series of timestamps $t_1 < t_2 < \ldots < t_s$ for $s = \mathcal{O}\left(\log n\right)$, such that $L_2(t_1 : m) > L_2(t_2 : m) > \ldots > L_2(t_s : m)$ and $t_1 \leq m - W + 1 < t_2$. Hence, $L_2(t_1 : m)$ is a constant factor approximation to $L_2(m - W + 1 : m)$, which is the $L_2$ norm of the sliding window. For each timestamp $t_i$ with $i \in [s]$, we also run an $L_2$-heavy hitter algorithm COUNTSKETCH$_i$, which outputs a list $\mathcal{L}_i$ of size at most $\mathcal{O}\left(\frac{1}{\alpha^2}\right)$ that contains the $L_2$-heavy hitters of the suffix of the stream starting at time $t_i$, as well as approximations to each of their frequencies. It might be tempting to simply output a noisy version of the list $\mathcal{L}_1$ output by COUNTSKETCH$_1$, since $t_1$ and $t_2$ sandwich the start of the sliding window, $m - W + 1$. Indeed, this is the approach by (Upadhyay, 2019), although they only consider the $L_1$-heavy hitter algorithm COUNTMIN because they study the weaker $L_1$-heavy hitter problem and they do not need to run a norm estimation algorithm because $L_1$ can be computed exactly. However, (Braverman et al., 2014; 2018) crucially note that $\mathcal{L}_1$ can also include a number of items that are heavy-hitters with respect to the suffix of the stream starting at time $t_1$ but *are not* heavy-hitters in the sliding window because many or even all of them appeared before time $m - W + 1$. Thus although $\mathcal{L}_1$ can guarantee that all the $L_2$-heavy hitters are reported by considering a lower threshold, say $\frac{\alpha}{2}$, the frequencies of each reported heavy-hitter can be arbitrarily inaccurate.

Observe it does not suffice to instead report the $L_2$-heavy hitters starting from time $t_2$. Although this will remove the false-positive issue of outputting items that are not heavy-hitters, there is now a false-negative issue; there may be heavy-hitters that appear after time $m - W + 1$ but before time $t_2$ that will not be detected by COUNTSKETCH$_2$. Hence, there may be heavy-hitters of the sliding window that are not reported by $\mathcal{L}_2$. See Figure 1 for an example.

**Approximate counters.** The fix by (Braverman et al., 2014; 2018) that is missed by (Upadhyay, 2019) is to run approximate counters for each item $k \in [n]$ reported by some heavy-hitter algorithm COUNTSKETCH$_i$, i.e., there exists $i \in [s]$ such that $k \in \mathcal{L}_i$. An approximate counter is simply a sliding window algorithm that reports a constant factor approximation to the frequency of a specific item $k \in [n]$. One way to achieve an approximate counter is to use the smooth histogram framework (Braverman & Ostrovsky, 2007), but we show that an improved accuracy can be guaranteed if the maintenance procedure instead considers additive error rather than multiplicative error. Given the approximate counter that reports an estimate $\widehat{f_k}$ as the frequency for an item $k \in [n]$, we can then compare $\widehat{f_k}$ to the estimated $L_2$ norm of the sliding window to determine whether $k$ could possibly be an $L_2$-heavy hitter. This rules out the false positives that can be returned in $\mathcal{L}_1$ without incurring false negatives omitted by $\mathcal{L}_2$.

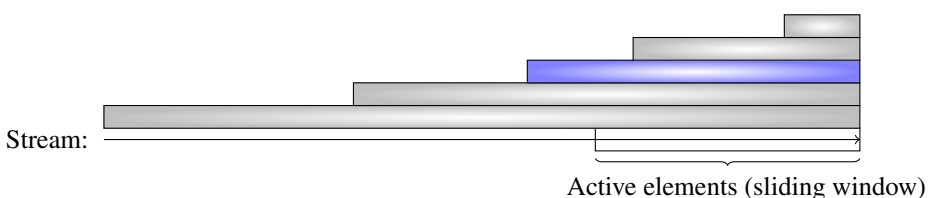

Stream:

Active elements (sliding window)

Fig. 1: Informally, we start a logarithmic number of streaming algorithms (the grey rectangles) at different points in time. We call the algorithm with the shortest substream that contains the active elements at the end of the stream (the blue rectangle). The challenge is that there may be heavy-hitters with respect to the blue rectangle that only appear before the active elements and therefore may be detected as heavy-hitters of the sliding window even though they are not.

**Large sensitivity of subroutines.** So far we have only discussed the techniques required to release $L_2$-heavy hitters in the non-DP setting. In order to achieve differential privacy, a first attempt might be to add Laplacian noise to each of the procedures. Namely, we would like to add Laplacian noise to the estimate of the $L_2$ norm of the sliding window and the frequency of each reported heavy-hitter. However, since both the estimate of the $L_2$ norm of the sliding window and the frequency of each reported heavy-hitter is governed by the timestamps $t_1, \ldots, t_s$, then the sensitivity of each quantity can be rather large. In fact, if the frequency of each reported heavy-hitter has sensitivity $\alpha \cdot L_2(m - W + 1 : m)$ through the approximate counters, then with high probability, the Laplacian noise added to the frequency of some reported heavy-hitter will completely dominate the actual frequency of the item to the point where it is no longer possible to identify the heavy-hitters. Thus the approximate counters missed by (Upadhyay, 2019) actually pose a significant barrier to the privacy analysis of the algorithm.

**Noisy timestamps.** A natural idea might be to make the timestamps in the histogram themselves noisy, e.g., by adding Laplacian noise to each of the timestamps. Unfortunately, we would no longer have sketches that correspond to the noisy timestamps in the sense that if the smooth histogram maintains a heavy-hitter algorithm COUNTSKETCH$_1$ starting at a time $t_1$ and prior to releasing the statistics, we add noise to the value of $t_1$ and obtain a noisy timestamp $\tilde{t_1}$, then we do not actually have a streaming algorithm starting at a time $\tilde{t_1}$.

**Lower smooth sensitivity through better approximations.** Instead, we guarantee differential privacy using the notion of smooth sensitivity (Nissim et al., 2007). The idea is the following — given an $\alpha$-approximation algorithm $\mathcal{A}$ for a function with sensitivity $\Delta_f$, we would like to intuitively say the approximation algorithm has sensitivity $\alpha \Delta_f$. Unfortunately, this is not true because $\mathcal{A}(X)$ may report $\alpha \cdot f(X)$ and $\mathcal{A}(Y)$ may report $\frac{1}{\alpha} \cdot f(Y)$ for adjacent datasets $X$ and $Y$. However, if $\mathcal{A}$ is instead a $(1 + \alpha)$-approximation algorithm, then difference of the output of $\mathcal{A}$ on $X$ and $Y$ can be bounded by $\alpha \cdot f(X) + \alpha \cdot f(Y) + \Delta_f$ through a simple triangle inequality, *conditioned on the correctness* of $\mathcal{A}$. In other words, if $\alpha$ is sufficiently small, then we can show that the *local sensitivity* of $\mathcal{A}$ is sufficiently small, which allows us to control the amount of Laplacian noise that must be added through existing mechanisms for smooth sensitivity. Unfortunately, if $\mathcal{A}$ is not correct, then even the local sensitivity could be quite large; we handle these cases separately by analyzing the smooth sensitivity of an approximation algorithm that is always correct and then arguing indistinguishability through statistical distance. Therefore, we can set the accuracy of the $L_2$ norm estimation algorithm, each $L_2$-heavy hitter algorithm, and each approximate counter algorithm to be sufficiently small and finally we can add Laplacian noise to each procedure without significantly impacting the final check of whether the estimated frequency for each item exceeds the heavy-hitter threshold.

**Pure differential privacy for $L_1$-heavy hitters in the sliding window model.** Due to the linearity of $L_1$, our algorithm for differentially private $L_1$-heavy hitters in the sliding window model is significantly simpler than the $L_2$-heavy hitters algorithm. For starters, each set of $c$ updates must contribute exactly $c$ to the $L_1$ norm, whereas their contribution to the $L_2$ norm depends on the particular coordinates they update. Therefore, not only do we not require an algorithm to approximate the $L_1$ norm of the active elements of the sliding window, but also we can fix a set of static timestamps in the smooth histogram, so we do not need to perform the same analysis to circumvent the sensitivity of the timestamps. Instead, it suffices to initialize a *deterministic $L_1$-heavy hitter* algorithm at each timestamp and maintain deterministic counters for each reported heavy-hitter. Pure differential

privacy then follows from the lack of failure conditions in the subroutines, which was not possible for $L_2$-heavy hitters.

## 2 PRELIMINARIES

For an integer $n > 0$, we use the notation $[n] := \{1, \ldots, n\}$. We use the notation $\text{poly}(n)$ to represent a constant degree polynomial in $n$ and we say an event occurs *with high probability* if the event holds with probability $1 - \frac{1}{\text{poly}(n)}$.

**Differential privacy.** In this section, we first introduce simple or well-known results from differential privacy. We say that streams $\mathfrak{S}$ and $\mathfrak{S}'$ are *neighboring*, if there exists a single update $i \in [m]$ such that $u_i \neq u_i'$, where $u_1, \ldots, u_m$ are the updates of $\mathfrak{S}$ and $u_1', \ldots, u_m'$ are the updates of $\mathfrak{S}'$.

**Definition 2.1** ($L_1$ sensitivity). *The $L_1$ sensitivity of a function $f : \mathcal{U}^* \to \mathbb{R}^k$ is defined by $\Delta_f = \max_{x,y \in \mathcal{U}^*|, \|x-y\|_1 = 1} \|f(x) - f(y)\|_1$.*

The $L_1$ sensitivity of a function $f$ bounds the amount that $f$ can change when a single coordinate of the input to $f$ changes and is often used to parameterize the amount of added noise to ensure differential privacy. We define the following notion of local $L_1$ sensitivity for a fixed input, which can be much smaller than the (global) $L_1$ sensitivity.

**Definition 2.2** (Local sensitivity). *For $f : \mathcal{U}^* \to \mathbb{R}$ and $x \in \mathcal{U}^*$, the local sensitivity of $f$ at $x$ is defined as $LS_f(x) = \max_{y:\|x-y\|_1=1} \|f(x) - f(y)\|_1$.*

Unfortunately, the local sensitivity can behave wildly for specific algorithms. Thus we have the following definition that smooths such behavior for local sensitivity.

**Definition 2.3** (Smooth upper bound on local sensitivity). *For $\beta > 0$, a function $S : \mathcal{U}^* \to \mathbb{R}$ is a $\beta$-smooth upper bound on the local sensitivity of $f : \mathcal{U}^* \to \mathbb{R}$ if*

*(1) For all $x \in \mathcal{U}^*$, we have $S(x) \geq LS_f(x)$.*

*(2) For all $x, y \in \mathcal{U}^*$ with $\|x - y\|_1 = 1$, we have $S(x) \leq e^\beta \cdot S(y)$.*

Even though the local sensitivity can be much smaller than the global $L_1$ sensitivity, the Laplace mechanism adds noise scaling with the global $L_1$ sensitivity. Hence it seems natural to hope for a mechanism that adds less noise. The following result shows that this is indeed possible.

**Theorem 2.4** (Corollary 2.4 in (Nissim et al., 2007)). *Let $f : \mathcal{U}^* \to \mathbb{R}$ and $S : \mathcal{U}^* \to \mathbb{R}$ be a $\beta$-smooth upper bound on the local sensitivity of $f$. If $\beta \leq \frac{\varepsilon}{2\ln(2/\delta)}$ and $\delta \in (0,1)$, then the mechanism that outputs $f(x) + X$, where $X \sim \mathsf{Lap}\left(\frac{2S(x)}{\varepsilon}\right)$ is $(\varepsilon, \delta')$-differentially private, for $\delta' = \frac{\delta}{2}\left(1 + \exp\left(\frac{\varepsilon}{2}\right)\right)$.*

**Heavy hitters.** We now formally introduce the $L_p$-heavy hitter problem and the algorithm COUNTSKETCH, which is commonly used to find the $L_2$-heavy hitters.

**Definition 2.5** ($L_p$-heavy hitter problem). *Given an accuracy/threshold parameter $\alpha \in (0,1)$, $p > 0$, and a frequency vector $f \in \mathbb{R}^n$, report all coordinates $k \in [n]$ such that $f_k \geq \alpha L_p(f)$ and no coordinates $j \in [n]$ such that $f_j \leq \frac{\alpha}{2} L_p(f)$. For each reported coordinate $k \in [n]$, also report an estimated frequency $\widehat{f}_k$ such that $|\widehat{f}_k - f_k| \leq \frac{\alpha}{4} L_p(f)$.*

**Theorem 2.6** (Heavy-hitter algorithm COUNTSKETCH (Charikar et al., 2004)). *Given an accuracy parameter $\alpha > 0$ and a failure probability $\delta \in (0,1)$, there exists a one-pass streaming algorithm COUNTSKETCH for the $L_2$-heavy hitter problem that uses $\mathcal{O}\left(\frac{1}{\alpha^2}\log\frac{n}{\delta}\right)$ words of space and $\mathcal{O}\left(\log\frac{n}{\delta}\right)$ update time.*

**Sliding window model.** In this section, we introduce simple or well-known results for the sliding window model.

**Definition 2.7** (Sliding window model). *Given a universe $\mathcal{U}$ of items, which we associate with $[n]$, let a stream $\mathfrak{S}$ of length $m$ consist of updates $u_1, \ldots, u_m$ to the universe $\mathcal{U}$, so that $u_i \in [n]$ for each $i \in [m]$. After the stream, a window parameter $W$ is given, which induces the frequency vector $f \in \mathbb{R}^n$ so that $f_k = |\{i : u_i = k \land i \geq m - W + 1\}|$ for each $k \in [n]$. In other words, each coordinate $k$ of the frequency vector is the number of updates to $k$ within the last $W$ updates.*

We say $A$ and $B$ are *adjacent* substreams of a stream $\mathfrak{S}$ of length $m$ if $A$ consists of the updates $u_i, \ldots, u_j$ and $B$ consists of the updates $u_{j+1}, \ldots, u_k$ for some $i, j, k \in [m]$. We have the following definition of a smooth function for the purposes of sliding window algorithms, not to be confused with the smooth sensitivity definition for differential privacy.

**Definition 2.8** (Smooth function). *Given adjacent substreams $A$ and $B$, a function $g : \mathcal{U}^* \to \mathbb{R}$ is $(\alpha, \beta)$-smooth if $(1 - \beta)g(A \cup B) \leq g(B)$ implies $(1 - \alpha)g(A \cup B \cup C) \leq g(B \cup C)$ for some parameters $0 < \beta \leq \alpha < 1$ and any adjacent substream $C$.*

Smooth functions are a key building block in the smooth histogram framework by (Braverman & Ostrovsky, 2010), which creates a sliding window algorithm for a large number of functions using multiple instances of streaming algorithms starting at different points in time. See Algorithm 1 for more details on the smooth histogram.

**Theorem 2.9** (Smooth histogram (Braverman & Ostrovsky, 2010)). *Given accuracy parameter $\alpha \in (0, 1)$, failure probability $\delta \in (0, 1)$ and an $(\alpha, \beta)$-smooth function $g : \mathcal{U}^m \to \mathbb{R}$, suppose there exists an insertion-only streaming algorithm $\mathcal{A}$ that outputs a $(1 + \alpha)$-approximation to $g$ with high probability using space $\mathcal{S}(\alpha, \delta, m, n)$ and update time $\mathcal{T}(\alpha, \delta, m, n)$. Then there exists a sliding window algorithm that outputs a $(1 + \alpha)$-approximation to $g$ with high probability using space $\mathcal{O}\left(\frac{1}{\beta}(\mathcal{S}(\beta, \delta, m, n) + \log m)\log m\right)$ and update time $\mathcal{O}\left(\frac{1}{\beta}(\mathcal{T}(\beta, \delta, m, n))\log m\right)$.*

---

**Algorithm 1** Smooth histogram (Braverman & Ostrovsky, 2010)

---

**Input:** Stream $\mathfrak{S}$, accuracy parameter $\rho \in (0, 1)$, streaming algorithm $\mathcal{A}$ for $(\rho, \beta(\rho))$-smooth function
**Output:** $(1 + \rho)$-approximation of predetermined function with probability at least $1 - \delta$
1: $H \leftarrow \emptyset$
2: **for** each update $u_t$ with $t \in [m]$ **do**
3:      $H \leftarrow H \cup \{t\}$
4:      **for** each time $t_s \in H$ **do**
5:          Let $x_s$ be the output of $\mathcal{A}$ with failure probability $\frac{\delta}{\text{poly}(n,m)}$ starting at time $t_s$ and ending at time $t$.
6:          **if** $x_{s-1} \leq \left(1 - \frac{\beta(\rho)}{2}\right)x_{s+1}$ **then**
7:              Delete $t_s$ from $H$ and reorder the indices in $H$
8: Let $s$ be the smallest index such that $t_s \in H$ and $t_s \leq m - W + 1$.
9: Let $x_s$ be the output of $\mathcal{A}$ starting at time $t_s$ at time $t$.
10: **return** $x_s$

---

We slightly tweak the smooth histogram framework to achieve a deterministic algorithm COUNTER that can be parametrized to give an additive $M$-approximation to the estimated frequency $\widehat{f_i}$ of a particular element $i \in [n]$ in the sliding window model.

**Lemma 2.10.** *There exists a deterministic algorithm COUNTER that outputs an additive $M$ approximation to the frequency of an element $i \in [n]$ in the sliding window model. The algorithm uses $\mathcal{O}\left(\frac{f_i}{M}\log m\right)$ bits of space.*

## 3 DIFFERENTIALLY PRIVATE HEAVY-HITTERS IN THE SLIDING WINDOW MODEL

In this section, we give a private algorithm for $L_2$-heavy hitters in the sliding window model. Our algorithm will initially use a smooth histogram approach by instantiating a number of $L_2$ norm estimation algorithm starting at various timestamps in the stream. Through a sandwiching argument, these $L_2$ norm estimation algorithms will provide a constant factor approximation to the $L_2$ norm of the sliding window, which will ultimately allow us to determine whether elements of the stream are heavy-hitters. Moreover, by using a somewhat standard smooth sensitivity argument, we can show that these subroutines can be maintained in a way that preserves differential privacy.

To identify a subset of elements that can be heavy-hitters, we also run a private $L_2$-heavy hitters algorithm starting at each timestamp. Unfortunately, because the timestamps do not necessarily

coincide with the beginning of the sliding window, it may be possible that depending on our approach, we may either output a number of elements with very low, possibly even zero, frequency, or we may neglect to output a number of heavy-hitters. To overcome this issue, we maintain a private algorithm COUNTER that outputs an estimated frequency for each item that is reported by our private $L_2$-heavy hitters algorithms, which allows us to rule out initially reported false positives without incurring false negatives. We give the algorithm in full in Algorithm 2.

---

**Algorithm 2** Differentially private sliding window algorithm for $L_2$-heavy hitters

---

**Input:** Stream $\mathfrak{S}$, accuracy parameter $\alpha \in (0,1)$, differential privacy parameters $\varepsilon, \delta > 0$, window parameter $W > 0$, size $n$ of the underlying universe, upper bound $m$ on the stream length
**Output:** A list $\mathcal{L}$ of $L_2$-heavy hitters with approximate frequencies
1: Process the stream $\mathfrak{S}$, maintaining timestamps $t_1, \ldots, t_s$ at each time $t \in [m]$ so that for each $i \in [s]$, either $i = s$, $t_{i+1} = t_i + 1$ or $L_2(t_i, t) \leq \left(1 + \left(\frac{\varepsilon}{1000 \log m}\right)^2\right) L_2(t_{i+1}, t)$ through a smooth histogram with failure probability $\frac{\delta}{2m^2}$
2: Implement heavy-hitter algorithm COUNTSKETCH on the substream starting at $t_i$ for each $i \in [s]$ with threshold $\frac{\alpha^3 \varepsilon}{500 \log m}$ and failure probability $\frac{\delta}{2m^2}$
3: Set $a = \max\{i \in [s] : t_i \leq m - W + 1\}$ on window query $W > 0$
4: Set $\widehat{L_2}$ to be an $\left(1 + \frac{\varepsilon}{500 \log m}\right)$-approximation to $L_2(t_a, t)$ from the smooth histogram and $X \leftarrow \mathsf{Lap}\left(\frac{1}{40 \log m} \widehat{L_2}\right)$
5: **for** each heavy-hitter $k \in [n]$ reported by COUNTSKETCH starting at $t_a$ **do**
6:     Run COUNTER with additive error $\frac{\alpha^3 \varepsilon}{1000 \log m} \widehat{L_2}$ for each reported heavy-hitter
7:     Let $\widehat{f_k}$ be the approximate frequency reported by COUNTER
8:     $Y_k \leftarrow \mathsf{Lap}\left(\frac{\alpha}{75 \log m} \widehat{L_2}\right)$, $Z_k \leftarrow \mathsf{Lap}\left(\frac{\alpha}{75 \log m} \widehat{L_2}\right)$, $\widetilde{f_k} = \widehat{f_k} + Z_k$
9:     **if** $\widetilde{f_k} \geq \frac{3\alpha}{4}\left(\widehat{L_2} + X\right) + Y_k$ **then**
10:         $\mathcal{L} \leftarrow \mathcal{L} \cup \{(k, \widetilde{f_k})\}$
11: **return** $\mathcal{L}$

---

We first describe the procedure for the approximate frequency estimation for each reported heavy-hitter. Let COUNTSKETCH$_a$ be an $L_2$-heavy hitter algorithm starting at timestamp $t_a$, where $a = \max\{i \in [s] : t_i \leq m - W + 1\}$ on window query $W > 0$. For a coordinate $k \in [n]$ that is reported by COUNTSKETCH$_a$ from times $t$ through $m$, we use COUNTER to maintain a number of timestamps such that the frequency of $k$ on the suffixes induced by the timestamps are arithmetically increasing by roughly $\alpha^2 L_2(f)/16$. We emphasize that we run the COUNTER for each reported heavy-hitter in the same pass as the rest of the algorithm.

**Lemma 3.1.** *Let $\mathcal{E}$ be the event that (1) the smooth histogram data structure does not fail, (2) all instances of COUNTSKETCH do not fail, and (3) $X \leq \frac{L_2(f)}{10}$ and $\max_{j \in [n]}(Y_j, Z_j) \leq \frac{\alpha L_2(f)}{10}$. Let COUNTSKETCH$_a$ be the instance of COUNTSKETCH starting at time $t_a$. Conditioned on $\mathcal{E}$, then for each reported heavy-hitter $k$ by COUNTSKETCH$_a$, Algorithm 2 outputs an estimated frequency $\widehat{f_k}$ such that $|f_k - \widehat{f_k}| \leq \frac{\alpha^3 \varepsilon}{500 \log n} L_2(f)$. The algorithm uses $\mathcal{O}\left(\frac{1}{\alpha^6 \varepsilon^2} \log^3 m\right)$ space and $\mathcal{O}\left(\frac{\log^2 m}{\alpha^4 \varepsilon^2}\right)$ update time per instance of COUNTSKETCH.*

We first show that the list $\mathcal{L}$ output by Algorithm 2 does not contain any items with "low" frequency.

**Lemma 3.2** (Low frequency items are not reported)**.** *Let $\mathcal{E}$ be the event that (1) the smooth histogram data structure does not fail, (2) all instances of COUNTSKETCH do not fail, and (3) $X \leq \frac{L_2(f)}{10}$ and $\max_{j \in [n]}(Y_j, Z_j) \leq \frac{\alpha L_2(f)}{10}$. Let $f$ be the frequency vector induced by the sliding window parameter $W$ and suppose $f_k \leq \frac{\alpha}{2} L_2(f)$. Then conditioned on $\mathcal{E}$, $k \notin \mathcal{L}$.*

We then show that the heavy-hitters are reported and bound the error in the estimated frequency for each reported item.

**Lemma 3.3** (Heavy-hitters are estimated accurately)**.** *Let $f$ be the frequency vector induced by the sliding window parameter $W$. Let $\mathcal{E}$ be the event that (1) the smooth histogram data structure does*

*not fail, (2) all instances of* COUNTSKETCH *do not fail, and (3)* $X \leq \frac{L_2(f)}{10}$ *and* $\max_{j \in [n]}(Y_j, Z_j) \leq \frac{\alpha L_2(f)}{10}$. *Conditioned on* $\mathcal{E}$, *then* $k \in \mathcal{L}$ *for each* $k \in [n]$ *with* $f_k \geq \alpha L_2(f)$. *Moreover, for each item* $k \in \mathcal{L}$, $|f_k - \widehat{f_k}| \leq \frac{\alpha^3 \varepsilon}{500 \log m} L_2(f)$.

We show that the event $\mathcal{E}$ conditioned by Lemma 3.1, Lemma 3.2, and Lemma 3.3 occurs with high probability.

**Lemma 3.4.** *Let* $\mathcal{E}$ *be the event that (1) the smooth histogram data structure does not fail on either stream, (2) all instances of* COUNTSKETCH *do not fail, and (3)* $X \leq \frac{L_2(f)}{10}$ *and* $\max_{j \in [n]}(Y_j, Z_j) \leq \frac{\alpha L_2(f)}{10}$. *Then* $\mathbf{Pr}\left[\mathcal{E}\right] \geq 1 - \frac{4}{m^2} - \frac{2}{m^{\frac{11}{4}}}$.

Before analyzing the privacy guarantees of Algorithm 2, we must analyze the local sensitivity of its subroutines. We first show a $\beta$-smooth upper bound on the local sensitivity of the frequency moment. We defer this statement to the supplementary material and also show the similar following $\beta$-smooth upper bound on the local sensitivity for each estimated frequency output by Algorithm 2.

**Lemma 3.5** (Smooth sensitivity of the estimated frequency). *Let* $\mathfrak{S}$ *be a data stream of length* $m$ *that induces a frequency vector* $f$ *and let* $\widehat{f_k}$ *be the estimate of the frequency of a coordinate* $k \in [n]$ *output by the smooth histogram. Define the function* $h(f)$ *by*

$$h(f) = \begin{cases} \widehat{f_k}, & \text{if } f_k - \frac{\alpha^3 \varepsilon}{1000 \log m} L_2(f) \leq \widehat{f_k} \leq f_k + \frac{\alpha^3 \varepsilon}{1000 \log m} L_2(f), \\ f_k - \frac{\alpha^3 \varepsilon}{1000 \log m} L_2(f), & \text{if } \widehat{f_k} < f_k - \frac{\alpha^3 \varepsilon}{1000 \log m} L_2(f), \text{ and} \\ f_k + \frac{\alpha^3 \varepsilon}{1000 \log m} L_2(f), & \text{if } \widehat{f_k} > f_k + \frac{\alpha^3 \varepsilon}{1000 \log m} L_2(f). \end{cases}$$

*Then the function* $S(f) = \frac{\alpha^3 \varepsilon}{200 \log m} h(f) + 2$ *is a* $\beta$-*smooth upper bound on the local sensitivity of* $h(f)$ *for* $\beta \geq \frac{\alpha^3 \varepsilon}{150 \log m}$, $\varepsilon > \frac{1000 \log m}{\sqrt{W} \alpha^3}$, *and sufficiently large* $W$.

With the structural results on smooth sensitivity in place, we show that Algorithm 2 is $(\varepsilon, \delta)$-differentially private.

**Lemma 3.6.** *There exists an algorithm (see Algorithm 2) that is* $(\varepsilon, \delta)$-*differentially private for* $\alpha \in (0, 1)$, $\varepsilon > \frac{1000 \log m}{\sqrt{W} \alpha^3}$, *and* $\delta > \frac{6}{m^2}$.

Since Algorithm 2 further adds Laplacian noise $Z \sim \mathsf{Lap}\left(\frac{\alpha}{75 \log m} \widehat{L_2}(f)\right)$ to each $\widehat{f_k}$ with $k \in \mathcal{L}$, then Lemma 3.3 implies that the additive error to $f_k$ is $\frac{\alpha}{50 \log m} L_2(f) + \mathsf{Lap}\left(\frac{\alpha}{75 \log m} \widehat{L_2}(f)\right)$ for each reported coordinate $k \in [n]$.

Thus through Lemma 3.6 (privacy), Lemma 3.2 and Lemma 3.3 (heavy-hitters/accuracy), and a simple analysis for space complexity, we have Theorem 1.2, our main result for differentially private $L_2$-heavy hitters in the sliding window model.

**Pure differential privacy for $L_1$-heavy hitters and continual release for $L_2$-heavy hitters in the sliding window model.** To achieve pure differential privacy, we use a deterministic $L_1$-heavy hitter algorithm MISRAGRIES at each timestamp and maintain deterministic counters for each reported heavy-hitter. Due to the linearity of $L_1$, the global $L_1$ sensitivity of our algorithm is at most 2 and thus it suffices to use the Laplace mechanism to guarantee pure differential privacy.

To achieve continual release of $L_2$-heavy hitters in the sliding window model, our algorithm consists of $L := \mathcal{O}\left(\log W\right) = \mathcal{O}\left(\log n\right)$ levels of subroutines. In each level $\ell \in [L]$, we split the stream into continuous blocks of length $S_\ell := 2^{\ell-2} \cdot \frac{\alpha \sqrt{W}}{100 \log W}$. Given a threshold parameter $\alpha > 0$, for each block in level $\ell$, we run MISRAGRIES with threshold $\frac{1}{2^{\ell+1} L}$. At the end of the stream, we stitch together a sketch of the underlying dataset represented by the sliding window through a binary tree mechanism. Due to a sharper balancing argument and analysis than previous work for continual release of $L_1$-heavy hitters, we obtain more accurate estimates of each item, which translates to sufficiently small error to catch the $L_2$-heavy hitters. We defer full details of both procedures to the supplementary materials.

## Acknowledgements

We would like to thank Sofya Raskhodnikova for clarifying discussions about smooth sensitivity. Jeremiah Blocki was supported in part by NSF CCF-1910659, NSF CNS-1931443, and NSF CAREER award CNS-2047272. Seunghoon Lee was supported by NSF CAREER award CNS-2047272. Tamalika Mukherjee was supported in part by Purdue Bilsland Dissertation Fellowship, NSF CCF-1910659, and NSF CCF-2228814. Work done in part while Samson Zhou was at Carnegie Mellon University and supported by a Simons Investigator Award of David P. Woodruff and by the National Science Foundation under Grant No. CCF-1815840.

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
