# OpenReview forum: "Differentially Private $L_2$-Heavy Hitters in the Sliding Window Model"
_ICLR.cc/2023/Conference — ICLR 2023 notable top 25%_

### Official Review · Reviewer_eT6L · 2022-10-22

**Confidence:** 3
**Correctness:** 3
**Technical Novelty And Significance:** 4
**Empirical Novelty And Significance:** Not applicable
**Recommendation:** 8

**Clarity, Quality, Novelty And Reproducibility:**

The write-up can be unclear in some places. One issue that's not explained carefully is why the instances of the Counter algorithm can be run in the same pass as the rest of the algorithm. The issue is that the CountSketch will report heavy hitters at different time steps, so are the Counter-s started at these different time steps? Why is that not an issue for approximating the frequencies of these elements accurately? Presumably because they were light before the Count Sketch identified them as heavy? I did not find a clear explanation of this in the main submission, and also not in supplementary material after skimming it, although I may have missed something.

**Strength And Weaknesses:**

Private streaming algorithms on sliding windows may be a bit of a niche area, but I think the problems are interesting and reasonably well motivated. The quantitative bounds on the accuracy $\alpha$ and on the space complexity are probably not tight, and the dependence on several parameters is quite bad. Still, I consider the techniques interesting, and, as already mentioned, the problems are natural and motivated.

**Summary Of The Paper:**

This paper considers differentially private algorithms for finding heavy hitters over a sliding window in the streaming model. I.e., given a stream of updates, and a parameter $W$, the goal is to be able to compute the elements whose frequency dominates the last $W$ updates of the stream. In particular, the goal is compute the elements whose frequency is an $\alpha$-fraction of the $L_2$ norm of the frequency vector of suffix of the stream of length $W$. This is already a non-trivial problem without privacy constraints, and is even more challenging if we require the algorithm to satisfy differential privacy.

The main result of the paper is an algorithm for the problem above that uses the smooth histogram framework from streaming algorithms, and the smooth sensitivity method from differential privacy (there is no connection between the two notions of "smooth" here). The approximation $\alpha$ is on the order of $\left(\frac{\log m}{\varepsilon \sqrt{W}}\right)^{1/3}$, where $m$ is the length of the stream.

**Summary Of The Review:**

The paper gives guarantees for the natural problem of privately computing heavy hitters over a sliding window, and does so using an interesting combination of techniques. The results are likely not tight, and the write-up can be polished further.

---

> ### Author Response · Authors · 2022-11-08
> **Response to Reviewer eT6L**
>
> > The write-up can be unclear in some places. One issue that's not explained carefully is why the instances of the Counter algorithm can be run in the same pass as the rest of the algorithm. The issue is that the CountSketch will report heavy hitters at different time steps, so are the Counter-s started at these different time steps? Why is that not an issue for approximating the frequencies of these elements accurately? Presumably because they were light before the Count Sketch identified them as heavy? I did not find a clear explanation of this in the main submission, and also not in supplementary material after skimming it, although I may have missed something.
>
> Yes, the crucial observation is that because all elements that are not reported by CountSketch are light, then a heavy-hitter must first be reported by CountSketch. Although the Counter subroutine for a universe element i is initiated only after the CountSketch reports element i as being heavy, the Counter subroutine only misses a small number of the instances of i (because we run CountSketch with a significantly lower threshold). Hence the Counter subroutine will still approximate the frequencies of the elements reported by CountSketch with an accuracy that is enough to differentiate whether an element is heavy with respect to the sliding window.
>
> We described this intuition in the first three paragraphs of the technical overview in Section 1.3 and in particular the paragraph "Approximate counters". We also briefly mention this idea at the beginning of Section 3 of the supplementary material. However, we agree with the reviewer that the description in the main submission is perhaps omitting the intuition that although some elements are missed by the Counter subroutine, these elements do not greatly affect the overall accuracy because they are small in magnitude.
>
> Thanks for your feedback. We have added this discussion to Section 3 of the full version as well as additional exposition to provide more intuition and further improve overall presentation.

---

### Official Review · Reviewer_1XJv · 2022-10-23

**Confidence:** 3
**Correctness:** 3
**Technical Novelty And Significance:** 3
**Empirical Novelty And Significance:** Not applicable
**Recommendation:** 8

**Clarity, Quality, Novelty And Reproducibility:**

# Clarity:

1. You also mentioned \ell_1-heavy hitters in “our contributions” section. I don’t really get the improvement/context. Please clarify. In general, why do we care much about the \ell_1 case if the paper is mostly about \ell_2 (and I find it a bit confusing)?
2. In Sec 1.3, you used notations “L_2(t_1 : m)” — I guess it means the L_2 norm of the frequency vector defined with the items arriving between time step [t_1, t_1 + m). But I cannot find where this is defined in the discussion.
3. Page 6, the paragraph “Sliding window model”, you started with “in this section”, which sounds weird since it is already in the middle of the section
4. A minor comment: Algorithm 2 has inconsistent styles — “Set” is in capitalized, while “for/if” is not.

# Quality:

This is a theoretically solid paper. However, considering the audience in ICLR, it may be better to also provide an empirical study.

# Originality:

The \ell_2-heavy hitter is a central problem in the analysis of the frequency vector. The study of the problem in the sliding window is well motivated, and given the context, I find it timely. Even though the paper uses previous frameworks, but I find making them to fit nontrivial, which requires a certain level of technical novelty.


**Strength And Weaknesses:**

# Strength:

The paper is technically nontrivial. It starts with the smooth histogram framework which is standard in designing (streaming) algorithms in the sliding window setting. However, it is observed that the \ell_1-sensitivity, which is a crucial complexity measure for differential privacy, may be unbounded, and thus the naive Laplacian mechanism cannot be applied. To this end, the smoothed sensitivity framework (Nissim et al., 2007) is applied, and I find the analysis of the smoothed sensitivity for (a certain implementation of) smooth histogram framework interesting, and it may be of independent interest.

# Weakness:

There is no empirical evaluation. For instance, it may be interesting to see how the privacy hurts the accuracy, by comparing with the non-private \ell_2-heavy hitter algorithms.


**Summary Of The Paper:**

This paper gives a streaming algorithm for \ell_2 heavy-hitter problem in the sliding window setting. In particular, for frequency threshold \alpha, privacy parameter \epsilon, stream length m, the algorithm uses poly(eps^{-1} \alpha^{-1} log m) space, to report a list of coordinates that have real frequency between \alpha / 2 L_2(f) to \alpha L_2(f), where f is the frequency vector. Furthermore,  it also gives a point-wise frequency estimation with additive error \alpha /4 L_2(f) for the reported coordinates. This algorithm only generates output at the end of the stream, hence is the so-called “one-shot” algorithm. As a secondary result, the stronger continual release algorithm is also obtained, with point-wise additive error \alpha \sqrt{W} / 2, where W is the size of the sliding window.


**Summary Of The Review:**

The lack of empirical evaluation is a weakness, but I find the theoretical contribution solid, and I wish to recommend for acceptance.

---

> ### Author Response · Authors · 2022-11-08
> **Response to Reviewer 1XJv**
>
> > There is no empirical evaluation. For instance, it may be interesting to see how the privacy hurts the accuracy, by comparing with the non-private \ell_2-heavy hitter algorithms.
>
> The goal of our paper was primarily to develop private, accurate, and efficient heavy-hitter algorithms with worst-case guarantees in the sliding window model, as well as correct fatal flaws missed by previous works. We believe that empirical evaluation to confirm accuracy and efficiency would be valuable as future work.
>
> > You also mentioned \ell_1-heavy hitters in “our contributions” section. I don’t really get the improvement/context. Please clarify. In general, why do we care much about the \ell_1 case if the paper is mostly about \ell_2 (and I find it a bit confusing)?
>
> For $L_2$-heavy hitters, we achieve the first algorithms for the sliding window model for approximate differential privacy, i.e., with a small tunable failure probability $\delta$. Surprisingly, our techniques can be easily adapted to handle $L_1$-heavy hitters with pure differential privacy, i.e., $\delta=0$, which is significant because such a result was previously incorrectly claimed by [Upa19]. Thus the $L_1$ case not only corrects previous shortcomings, but also shows the generality of our techniques.
>
> [Upa19] Jalaj Upadhyay: Sublinear Space Private Algorithms Under the Sliding Window Model. ICML 2019: 6363-6372
>
> > In Sec 1.3, you used notations “L_2(t_1 : m)” — I guess it means the L_2 norm of the frequency vector defined with the items arriving between time step [t_1, t_1 + m). But I cannot find where this is defined in the discussion.
>
> The notation is defined at the end of Section 1.1 though we will slightly expand the discussion to improve clarity.
>
> > Page 6, the paragraph “Sliding window model”, you started with “in this section”, which sounds weird since it is already in the middle of the section
>
> Thanks, we have adjusted this language.
>
> > A minor comment: Algorithm 2 has inconsistent styles — “Set” is in capitalized, while “for/if” is not.
>
> Thanks, we have changed the style of Algorithm 2 to match that of Algorithm 1 (and more generally, propagated these changes across the other algorithms in the full version).

---

### Official Review · Reviewer_jYg8 · 2022-10-26

**Confidence:** 2
**Clarity, Quality, Novelty And Reproducibility:** See "Strength And Weaknesses" section.
**Correctness:** 3
**Technical Novelty And Significance:** 2
**Empirical Novelty And Significance:** Not applicable
**Recommendation:** 5

**Strength And Weaknesses:**

Strength:
1. The $L_2$-heavy hitter problem in the sliding window model is important.
2. A sublinear space algorithm under differential privacy is provided, which rigorous theoretical guarantees.

Weaknesses:
1. For the privacy guarantee, $\delta = O(\frac{1}{m^2})$ is somewhat weak. Often in differential privacy it is insisted that $\delta$ is cryptographically negligible.
2. The privacy guarantee is only for one-shot computation. I think in streaming data analysis especially under sliding window model, it is much more practical to consider online and adaptive data analysis.
3. The paper is quite dense, and it seems to me that the writing of the paper mainly aims for theory audience and it could be difficult for a typical ICLR audience to understand the main ideas.

**Summary Of The Paper:**

This paper studies the problem of how to compute $L_2$-heavy hitters in the sliding window model under differential privacy. The main contribution is giving the first differential private algorithm for this problem using sublinear working space.

**Summary Of The Review:**

The paper provides the first sublinear space algorithm for sliding window $L_2$-heavy hitters under differential privacy. However, the guarantee on privacy is weak and thus less interesting. The writing of the paper could also be improved to be more friendly to typical ICLR audience.

---

> ### Author Response · Authors · 2022-11-08
> **Response to Reviewer jYg8**
>
> > For the privacy guarantee, $\delta=O(\frac{1}{m^2})$ is somewhat weak. Often in differential privacy it is insisted that $\delta$ is cryptographically negligible.
>
> Our main theorem states that $\delta$ can be set to be $\frac{1}{m^c}$ for any constant $c>0$. Thus we achieve arbitrary inverse polynomial probability failure, i.e., significantly smaller failure probability than $\frac{1}{m^2}$. In fact, we remark that our results can be easily generalized to arbitrary $\delta$ at the cost of $\log\frac{1}{\delta}$ overhead in memory and runtime. This is a standard approach in decreasing the failure probability in streaming algorithms (which is the source of the privacy failure) -- we will add this discussion in the text.
>
> > The privacy guarantee is only for one-shot computation. I think in streaming data analysis especially under sliding window model, it is much more practical to consider online and adaptive data analysis.
>
> We emphasize that the full version of our paper in the supplementary material considers continual release for both $L_1$ and $L_2$ heavy-hitters. To the best of our knowledge, there is a lack of literature for adaptive output in the sliding window model, even for non-private algorithms. This would indeed be an interesting direction for future exploration, but we believe it is beyond the current scope of our paper. On the other hand, the online model is inherently orthogonal to the sliding window model because the online model generally seeks irrevocable decisions while the sliding window model generally deletes elements beyond a certain time, though please correct us if we have any misunderstandings of your suggestions.
>
> > The paper is quite dense, and it seems to me that the writing of the paper mainly aims for theory audience and it could be difficult for a typical ICLR audience to understand the main ideas.
>
> Thanks for the feedback. We will work to incorporate the reviewer feedback into the extended abstract to improve presentation. We also remark that the full version in the supplementary material contains considerably more exposition, providing both more background and more intuition. Finally, we point out that there is a strong presence of a theory-based differential privacy sub-community within ICLR and other machine learning conferences, e.g., consider the following papers since the beginning of last year.
>
> Lun Wang, Iosif Pinelis, Dawn Song: Differentially Private Fractional Frequency Moments Estimation with Polylogarithmic Space. ICLR 2022
>
> Hilal Asi, Vitaly Feldman, Kunal Talwar: Optimal Algorithms for Mean Estimation under Local Differential Privacy. ICML 2022: 1046-1056
>
> Jennifer Gillenwater, Matthew Joseph, Andres Munoz Medina, Monica Ribero Diaz: A Joint Exponential Mechanism For Differentially Private Top-k. ICML 2022: 7570-7582
>
> Haim Kaplan, Shachar Schnapp, Uri Stemmer: Differentially Private Approximate Quantiles. ICML 2022: 10751-10761
>
> Ziyue Huang, Yuting Liang, Ke Yi: Instance-optimal Mean Estimation Under Differential Privacy. NeurIPS 2021: 25993-26004
>
> Kunho Kim, Sivakanth Gopi, Janardhan Kulkarni, Sergey Yekhanin: Differentially Private n-gram Extraction. NeurIPS 2021: 5102-5111
>
> Xiyang Liu, Weihao Kong, Sham M. Kakade, Sewoong Oh: Robust and differentially private mean estimation. NeurIPS 2021: 3887-3901
>
> Sofya Raskhodnikova, Satchit Sivakumar, Adam D. Smith, Marika Swanberg: Differentially Private Sampling from Distributions. NeurIPS 2021: 28983-28994
>
> Jennifer Gillenwater, Matthew Joseph, Alex Kulesza: Differentially Private Quantiles. ICML 2021: 3713-3722
>
> Dung Nguyen, Anil Vullikanti: Differentially Private Densest Subgraph Detection. ICML 2021: 8140-8151
>
> Gang Qiao, Weijie J. Su, Li Zhang: Oneshot Differentially Private Top-k Selection. ICML 2021: 8672-8681
>
> We believe that we have addressed your questions about the range of $\delta$, generalizations to continual release, and overall fit with ICLR. If you feel your concerns have indeed been addressed, we hope you will consider raising your score. Otherwise, we would also be happy to clarify any misunderstandings we may have about your concerns.

---

### Official Review · Reviewer_EhqA · 2022-10-27

**Confidence:** 3
**Clarity, Quality, Novelty And Reproducibility:** Good.
**Correctness:** 4
**Technical Novelty And Significance:** 3
**Empirical Novelty And Significance:** Not applicable
**Recommendation:** 5

**Strength And Weaknesses:**

I would consider the problem important, but perhaps a little specialized. Technically, the paper seems to put together several ingredients from the literature on DP and heavy hitters in the right way.  On one hand, I think it is above the bar technically. On the other hand, I cannot put my finger on any one ingredient which strikes me as particularly novel, it appears that the contribution is to understand the literature well, select the right tools and put them together in the right way,.


My main concern about the result is that as I understand it, the privacy parameter $\epsilon$ is rather large. The theorem states that $\epsilon > log(m)/(\alpha^3 \sqrt{W})$. It seems that if the window size needs to be at least $(log(m))^2/\alpha^6$ to have reasonable privacy guarantees.  This is followed by a comment about general $\alpha, \epsilon$ which did not make sense to me.
1.  They say that they allow $\alpha \geq 1$. I thought $\alpha$ is the fraction of the two norm $\|f\|_2$ coming from $f_i$, so having it greater than $1$ seems to not parse.
2. For general $\epsilon$, they say something about additive error in the utility. I am not sure if they are referring to (2) which is the heavy hitters guarantee, or (3) which is the accuracy guarantee. For how small an $\epsilon$ can the algorithm guarantee  the right list of heavy hitters?

It seems plausible that to get good estimates to the frequencies, one needs large $\epsilon$. Even for this, it seems that the bounds here are rather large.  What is less clear is to what extent getting the set of heavy hitters correct depends on the noise parameter.


**Summary Of The Paper:**

This paper considers the problem of finding $L_2$ heavy hitters in the sliding window model with differential privacy. If we relax either the privacy or the sliding window requirement, there were efficient algorithms but not with both.
It gives an efficient algorithm for this problem with polylog space.

**Summary Of The Review:**

The problem is interesting to the DP community, and potentially relevant in practice. My primary concern is about the restriction on $\epsilon$, which makes the result rather limited in scope. If there is an argument for why it should degrade for smaller windows or small $\alpha$, I might be more inclined to accept the paper. As it stands, I would lean to reject.

---

> ### Author Response · Authors · 2022-11-08
> **Response to Reviewer EhqA**
>
> > I would consider the problem important, but perhaps a little specialized...On the other hand, I cannot put my finger on any one ingredient which strikes me as particularly novel, it appears that the contribution is to understand the literature well, select the right tools and put them together in the right way.
>
> Given the data retention policies of many platforms, the private release of time-sensitive heavy-hitters is a natural question and hence there is precedence for the study of private heavy-hitters in the sliding window model, e.g., [Upa19]. We also remark that there has been a long line of literature studying the private release of heavy-hitters, e.g., [AS19, BNST17, HK12, CLSX12, KMSZ08]. Our work not only substantially improves upon, but also fixes subtle shortcomings in existing results.
>
> [AS19] Jayadev Acharya, Ziteng Sun: Communication Complexity in Locally Private Distribution Estimation and Heavy Hitters. ICML 2019: 51-60
>
> [BNST17] Raef Bassily, Kobbi Nissim, Uri Stemmer, Abhradeep Guha Thakurta: Practical Locally Private Heavy Hitters. NIPS 2017: 2288-2296
>
> [HK12] Justin Hsu, Sanjeev Khanna, Aaron Roth: Distributed Private Heavy Hitters. ICALP (1) 2012: 461-472
>
> [CLSX12] T.-H. Hubert Chan, Mingfei Li, Elaine Shi, Wenchang Xu: Differentially Private Continual Monitoring of Heavy Hitters from Distributed Streams. Privacy Enhancing Technologies 2012: 140-159
>
> [KMSZ08] Joe Kilian, Andre Madeira, Martin J. Strauss, Xuan Zheng: Fast Private Norm Estimation and Heavy Hitters. TCC 2008: 176-193
>
> [Upa19] Jalaj Upadhyay: Sublinear Space Private Algorithms Under the Sliding Window Model. ICML 2019: 6363-6372
>
> > My main concern about the result is that as I understand it, the privacy parameter $\epsilon$ is rather large. The theorem states that $\epsilon>\log(m)/(\alpha^3\sqrt{W})$. It seems that if the window size needs to be at least $(\log(m))^2/\alpha^6$ to have reasonable privacy guarantees.
>
> Since $W$ is the size of the window, it is also the size of the data. Thus it is reasonable to assume $W\gg\frac{\log^2 m}{\alpha^6}$ because otherwise, the entire dataset can be stored, and then the problem reduces to an offline release of private heavy-hitters rather than a streaming/sliding-window model.
>
> > They say that they allow $\alpha\ge 1$...I thought is the fraction of the two norms $|f|_2$ coming from $f_i$, so having it greater than seems to not parse...For general $\epsilon$, they say something about additive error in the utility. I am not sure if they are referring to (2) which is the heavy hitters guarantee, or (3) which is the accuracy guarantee. For how small an $\epsilon$ can the algorithm guarantee the right list of heavy hitters?
>
> The statement should say that we permit $\epsilon\ge 1$ due to setting $\eta=\max(1,\epsilon)$. For $\epsilon>\log(m)/(\alpha^3\sqrt{W})$, we achieve both the heavy-hitter guarantee and the accuracy guarantee claimed in the main theorem. For smaller $\epsilon$, we achieve an additive error guarantee in terms of $\epsilon$ for both the heavy-hitters guarantee and the accuracy guarantee. This is because for smaller $\epsilon$, the noise that must be added to each estimate is so large that it overwhelms the error allowed by any bounds that are independent of $\epsilon$. Regardless, we agree that these comments can be confusing and need not be there at all since we do not further elaborate and so we have removed these statements.
>
> > My primary concern is about the restriction on $\epsilon$, which makes the result rather limited in scope. If there is an argument for why it should degrade for smaller windows or small $\alpha$, I might be more inclined to accept the paper.
>
> Since the window size $W$ is the size of the data, the restriction that $\epsilon>\log(m)/(\alpha^3\sqrt{W})$ is quite mild. For example, the dataset [BDB16] referenced in our paper consists of 70 million entries that can be fully stored. Thus in practical settings, we should view $W$ as being significantly larger than 70 million for the sliding window model assumption that the dataset must be viewed as a data stream because it cannot be maintained as a single file.
>
> Mathematically speaking though, the reason the result degrades for smaller windows or small $\alpha$ is because the same privacy parameter becomes much relatively smaller compared to the data size, thereby requiring more relative noise to be added, to the point where the added noise is so large that it overwhelms any possible multiplicative guarantee.
>
> [BDB16] Jeremiah Blocki, Anupam Datta, Joseph Bonneau: Differentially Private Password Frequency Lists. NDSS 2016
>
> We hope the clarification that our algorithm allows for natural regimes of $\epsilon$ alleviates your concerns; if not, we would be happy to engage in a more thorough discussion. Otherwise, we hope that you will consider raising your score if there are no further concerns.

---

### Author Response · Authors · 2022-11-08
**Thanks to all reviewers**

We thank the reviewers for their valuable feedback. We are especially enheartened to see positive comments such as:
- I would consider the problem important...the problem is interesting to the DP community, and potentially relevant in practice (Reviewer EhqA)
- the paper...is above the bar technically (Reviewer EhqA)
- the $L_2$-heavy hitter problem in the sliding window model is important (Reviewer jYg8)
- A sublinear space algorithm under differential privacy is provided...rigorous theoretical guarantees (Reviewer jYg8)
- The \ell_2-heavy hitter is a central problem in the analysis of the frequency vector (Reviewer 1XJv)
- The paper is technically nontrivial...I find the analysis of the smoothed sensitivity for (a certain implementation of) smooth histogram framework interesting, and it may be of independent interest (Reviewer 1XJv)
- This is a theoretically solid paper...I find making [previous frameworks] to fit nontrivial, which requires a certain level of technical novelty (Reviewer 1XJv)
- I think the problems are interesting and reasonably well motivated (Reviewer eT6L)
- I consider the techniques interesting, and, as already mentioned, the problems are natural and motivated (Reviewer eT6L)
- The paper gives guarantees for the natural problem of privately computing heavy hitters over a sliding window, and does so using an interesting combination of techniques (Reviewer eT6L)

We have incorporated reviewer feedback into an updated version of the document to further improve overall presentation -- **we have uploaded this updated version as the rebuttal revision**. More generally, we provide our responses to the specific questions of each reviewer below. We hope our answers resolve all initial questions and concerns raised by the reviewers and we will be most happy to answer any remaining questions!

---

### Author Response · Authors · 2022-11-16
**Check for Questions?**

Hi everyone,

As the discussion period is drawing to a close, we wanted to check whether the reviewers (or the area chair) had any remaining unresolved questions or concerns that we could potentially address.

Thanks again for your consideration!

---

### Decision · Program_Chairs · 2023-01-20

**Decision:**

Accept: notable-top-25%

**Justification For Why Not Higher Score:**

I do not think the subject is of a lot of interest to the general audience.

**Justification For Why Not Lower Score:**

It's good to reward a well-crafted paper with at least a spotlight slot.


**Metareview: Summary, Strengths And Weaknesses:**

This paper addresses a recently standard program in DP analysis. It develops some new techniques for doing so, and the analysis is non-trivial.


**Note From Pc:**

if the above contains the word "oral" or "spotlight" please see: "oral" presentation means -> notable-top-5% and "spotlight" means -> notable-top-25%. As stated in our emails, we are disassociating presentation type from AC recommendations

**Summary Of Ac-Reviewer Meeting:**

I discussed with one reviewer the overall significance of the results in the context of prior work, and our impression was very positive.